# Noise Estimation for Image Sensor Based on Local Entropy and Median Absolute Deviation

**DOI:** 10.3390/s19020339

**Published:** 2019-01-16

**Authors:** Yongsong Li, Zhengzhou Li, Kai Wei, Weiqi Xiong, Jiangpeng Yu, Bo Qi

**Affiliations:** 1School of Microelectronics and Communication Engineering, Chongqing University, Chongqing 400044, China; liyongsong@cqu.edu.cn (Y.L.); kaiwei@cqu.edu.cn (K.W.); xiongweiqi_cqu@163.com (W.X.); jiangpengyu@cqu.edu.cn (J.Y.); 2Key Laboratory of Dependable Service Computing in Cyber Physical Society of Ministry of Education, Chongqing University, Chongqing 400044, China; 3Key Laboratory of Beam Control, Institute of Optics and Electronics, Chinese Academy of Sciences, Chengdu 610209, China; qibo@ioe.ac.cn

**Keywords:** noise estimation, additive white Gaussian noise, Poisson-Gaussian noise, local gray statistic entropy, local median absolute deviation, image sensor

## Abstract

Noise estimation for image sensor is a key technique in many image pre-processing applications such as blind de-noising. The existing noise estimation methods for additive white Gaussian noise (AWGN) and Poisson-Gaussian noise (PGN) may underestimate or overestimate the noise level in the situation of a heavy textured scene image. To cope with this problem, a novel homogenous block-based noise estimation method is proposed to calculate these noises in this paper. Initially, the noisy image is transformed into the map of local gray statistic entropy (LGSE), and the weakly textured image blocks can be selected with several biggest LGSE values in a descending order. Then, the Haar wavelet-based local median absolute deviation (HLMAD) is presented to compute the local variance of these selected homogenous blocks. After that, the noise parameters can be estimated accurately by applying the maximum likelihood estimation (MLE) to analyze the local mean and variance of selected blocks. Extensive experiments on synthesized noised images are induced and the experimental results show that the proposed method could not only more accurately estimate the noise of various scene images with different noise levels than the compared state-of-the-art methods, but also promote the performance of the blind de-noising algorithm.

## 1. Introduction

Images captured by charge-coupled device (CCD) image sensors will bring about many types of noise, which are derived from CCD image sensor arrays, camera electronics and analog to digital conversion circuits. Noise is the most concerning problem in imaging systems for it directly degrades the image quality. Therefore, image de-noising is a critical procedure to ensure high quality images. Most of existing de-noising algorithms process the noisy images via assuming that the noise level parameters are given. However, this is not the correct operation in real noisy images, because the noise level parameters generally are unknown in the real world. Blind de-noising without requesting any prior knowledge is a high-profile study. How to accurately estimate the noise parameters is a challenging issue in blind de-noising field, hence, noise level estimation has attracted many studies and a number of noise estimation algorithms have been developed [1,2,3].

Actually, estimating the CCD noise level function (NLF) is to extract ground true noise from noisy image by finding the proximal noise probability density function (PDF). Comprehensive noise PDF estimation may not be easy, yet in statistical science, if we know the model of distribution of the noise PDF, the numerical characteristics such as mathematical expectation, variance and covariance can explain the noise PDF credibly. The most prevalent noise model for CCD image sensors is the zero-mean additive white Gaussian noise (AWGN) model, and the goal of estimating AWGN noise PDF is to estimate one parameter, the standard deviation. In this field, many representative algorithms have been proposed and can be divided into four categories: CCD response-based methods, filter-based methods, statistics-based methods and block-based methods. In the CCD response-based methods, the image noise is assessed by noise sources, which are introduced by CCD cameras. In [4], according to a pair of studies on the CCD Photon Transfer Curve (PTC) and camera noise characters, a complete equation of different noise sources of image sensor was established. Irie et al. [5,6] introduced a CCD camera noise model that includes more comprehensive noise sources for CCD image sensors, and developed a robust technique to measure and distinguish those noise sources in CCD camera images. These methods are trying to separate the image noise one by one via analyzing the types of CCD sensor noise sources on the premise of acquiring abundant knowledge about the types of CCD image sensor noises and the procedures of measurement, but it is difficult to meet these needs. To address this problem, the non-linear camera response function (CRF) that transforms real scene light into an image pixel value is proposed in [7]. Furthermore, the algorithm depicted in [8] utilizes the pre-measured real world CRF to transform the CCD camera noise from light space into image domain, and then estimates the NLF by exploiting the piecewise smooth image noise model. These CCD response-based noise estimation methods can be further enhanced to study the sources of image sensor noise and identify the types of image noise. Nevertheless, the CCD response-based methods need more time to compute and require familiarity with the camera responses.

In the filter-based methods, the structures of image are suppressed with a high-pass filter, and then the noise variance of the filtered noisy image can be computed. In the early filter-based methods, Immerkær [9] adopted a fast Laplacian-based filter to strengthen the noise and reduce the structures, and estimated the noise standard deviation via averaging the whole filtered image. In order to further improve the accuracy of Immerkær’s estimation, Yang et al. [10] appended adaptive Sobel edge detector to eliminate edges before Laplacian filtering operation. A step signal filter model is presented in [11]. This model is a nonlinear combination of polarized and directional derivatives, and is used to address edges detection and noise estimation. These filter-based methods have outstanding performance in reducing the time and computational complexity. However, these methods are more likely to overestimate the noise level for rich textured images or low noise levels.

The statistics-based methods are based on observation and measurement of statistical values between noisy images and natural noise-free images and obtain the decisive relationship between noise standard deviation and those statistical values. Zoran and Weiss [12] reported that natural images have scale-invariant statistics and kurtosis values are higher for the low frequency component, and linked the discrete cosine transformation (DCT) kurtosis values to the noise standard deviation. Following DCT kurtosis noise estimation work, Lyu et al. [13] further extended their research to estimate local noise variance, and Dong et al. [14] developed a new kurtosis model using a K-means based method to estimate noise level. Actually, the DCT-based statistics methods work well for low-level noise, yet may underestimate the noise variance at a high noise level. In [15,16], the robust median absolute deviation (MAD) noise estimator suggested by Donoho et al., is one of the most commonly used noise estimation algorithms for its efficiency and accuracy.

Being different from the conventional work to process the whole image, the block-based methods try to select homogenous blocks without a high frequency component. Pyatykh et al. [17] analyzed the basis of principal component analysis (PCA) of image blocks and found that smallest eigenvalue of image blocks can be used to estimate noise variance. Liu et al. [18] suggested that low-rank can be suitable for weak-textured blocks selection, and then the noise level can be estimated from these selected blocks using iterative PCA. Recently, References [19,20] developed sparse representation models of selected homogenous blocks to recover NLF. References [21,22] designed superpixel-based noise estimation approaches by collecting irregular shaped smooth blocks. For their simplicity and efficiency, the block-based noise estimation methods are drawing more and more attention recently. Whereas, there is a challenge for stripping homogenous blocks that can easily influence the performance of estimation.

Some previous studies have mentioned that entropy can be used for image segmentation and texture analysis [23,24,25]. Meanwhile, other studies also reported that entropy has a robust stability to noise [26,27]. However, there is little research on applying the entropy to estimate image sensor noise. Considering the good performances of entropy, the local gray statistical entropy (LGSE) is proposed in this paper to extract blocks with weak textures. By making the utmost out of the block-based method and robust MAD-based method, a novel noise estimator including the LGSE for blocks selection and Haar wavelet-based local median absolute deviation (HLMAD) for local variance calculation is designed.

In our work, we also engage in Poisson-Gaussian noise (PGN), which is the less studied but better modeled for the actual camera noise. The existing PGN sensor noise estimators can be classified as: Scatterplot fitting-based methods, least square fitting-based methods and maximum likelihood estimation-based methods. The scatterplot fitting-based methods are proposed to group a set of local means and variances of the selected blocks, and to fit those scatter points using linear regression [28,29,30]. The least square fitting-based methods [31,32,33] use a least square approach to estimate PGN parameters by sampling weak textured blocks and fitting a group of sampled data. In [34,35], the maximum likelihood estimation-based methods still extract smooth blocks first, and then exploit the maximum likelihood function to estimate PGN parameters. The three categories of PGN estimators are equally popular for estimating PGN parameters. Specifically in this paper, we choose maximum likelihood estimation (MLE) to process extracted local mean-variance data for PGN estimation.

There are four contributions in this paper: (1) We analyze the noise sources of an imaging camera, and choose the simple but effective noise models in the image domain to describe most of noise forms of imaging sensor. (2) In the view of conventional noise estimation methods, textures and edges of noisy images are eliminated first in those algorithms. Inspired by this principle, an effective method using LGSE to select homogenous blocks without high frequency details is proposed in this paper. As far as we know, this is a novel approach to utilize local gray entropy in noise estimation. (3) By comparing traditional noise estimation methods, we find local median absolute deviation has a robust performance to compute local standard deviation and MLE can overcome the weakness of scatterplot-based estimation. By combining above two methods and their advantages, a reliable noise estimation method is established. (4) Due to these measures, a robust image sensor noise level estimation scheme is presented, and it is superior to some of the state-of-the-art noise estimation algorithms.

The structure of this paper is organized as follows: The noise model of image sensor in image domain is discussed, and the AWGN model and PGN model are presented in Section 2. The LGSE and HLMAD based noise estimation algorithm is described, and the whole details of the novel and robust method are presented in Section 3. Experimental results and discussions on synthesized noisy datasets are shown in Section 4. Finally, Section 5 gives the conclusion of this paper.

## 2. Image Sensor Noise Model

Irie et al. [5,6] indicted that the CCD camera noise model includes comprehensive noise sources: Reset noise, thermal noise, amplifier noise, flicker noise, circuit readout noise and photon shot noise. Here, the reset noise, thermal noise, amplifier noise, flicker noise and circuit readout noise vary only in additive spatial domain and can be generally included in the readout noise. Hence, the total noise of a CCD camera can be defined as the combination of the readout noise (additive Gaussian process) and photon shot noise (multiplicative Poisson process) [5].

In the CCD noise analysis experiment, we used a 532 nm He-Ne laser to generate light beams, chose CCD97 (produced by e2v, England) in the standard mode (without using multiplicative gain register) to realize the photo-electric conversion, and selected the 16-bit data acquisition card (0-65535 DN) to collect the images. In this experiment, the diaphragm was used to adjust the intensity of the illumination, and the polaroid and optical lens were used to preserve the illumination uniformity on CCD97 light receiving surface. In order to ensure the stability of the scene and the uniformity of the CCD97 received light, a photomultiplier tube was selected as the wide-range micro-illuminometer to monitor the stable brightness at the CCD light receiving surface before the experiment. In addition, the average operation of 25 images in each group can overcome the output fluctuation of He-Ne laser and promote the accuracy of system testing.

To identify and calculate the total noise, readout noise and shot noise of image sensor according to the measurements of [5,6], 100 image groups in different illumination intensities were captured in optical darkroom, and each group had 25 consecutive images without illumination and 25 consecutive images with illumination. Due to the captured images under uniform illumination intensity, the total noise variance in each group is equal to the average value of variances of 25 consecutive images with a given illumination. The variance of the readout noise in each group can be computed by averaging the variances of 25 images captured in dark conditions. The shot noise variance in each group is the difference value between total noise variance and readout noise variance. By measuring the noise variances of these images, the photon transfer curve (PTC) and its logarithm were computed. The experimental platform and the computed PTC curve are shown in Figure 1. Figure 1a shows the experimental platform of CCD noise analysis. Figure 1b depicts that the total noise of image sensor contains signal independent readout noise (red dots) and signal dependent shot noise (green dots). It can be seen from Figure 1b that the noise curve is almost flat under low illumination levels implying that the noise (mostly readout noise) in dark regions of an image is signal independent, while as the illumination increases, the shot noise becomes more and more signal dependent.

According to the noise analysis of the CCD image sensor, two parametric models were used to model the image sensor noise: The generalized zero-mean additive white Gaussian noise (AWGN) model and the Poisson-Gaussian noise (PGN) model.

### 2.1. Additive White Gaussian Noise Model

In the various sources of CCD camera noise, the reset noise, thermal noise, amplifier noise and circuit readout noise can be seen as additive spatial and temporal noise. For all of these noise sources that obey the zero-mean whited additive Gaussian distribution, their joint noise distribution model can be simplified as:(1)$$f(x,y)=f_{S}(x,y)+f_{N}(x,y),$$
where (x,y) is the pixel location, and f(x,y), $f_{S}(x,y)$, $f_{N}(x,y)$ denote original noisy image, noise-free image and random noise image at (x,y), respectively. Generally, the random noise image $f_{N}(x,y)$ is assumed to be a whited additive Gaussian distribution noise with zero mean and unknown variance $\sigma_{n}^{2}$:(2)$$f_{N}(x,y)\sim N(0,\sigma_{n}^{2}).$$

In the AWGN model, due to the images always having rich textures, the unknown variance $\sigma_{n}^{2}$ estimation still is a head-scratching issue for blind de-noising. To solve this problem, a novel noise variance estimation method is proposed in this paper.

### 2.2. Poisson-Gaussian Noise Model

For the practical image sensor noise model, the signal-dependent shot noise cannot be ignored. As depicted in Figure 1b, with the increase of irradiation, the measured shot noise also increased significantly. As the shot noise follows the Poisson distribution, the actual CCD camera noise model can be treated as a mixed model of multiplicative Poisson noise and additive Gaussian white noise:(3)$$f(x,y)=f_{S}(x,y)+\eta (x,y),$$
where η(x,y) is the mixed total noise image at (x,y). In the Poisson-Gaussian mixed noise model, the mixed total noise image η(x,y) contains signal-dependent (Poisson distribution) noise and signal-independent (Gaussian distribution) noise, and is modeled as:(4)$$\eta (x,y)=\rho p_{S}(x,y)+f_{N}(x,y),$$
where $p_{S}(x,y)$ follows Poisson distribution, that is $p_{S}(x,y)\sim P(f_{S}(x,y))$, and as mentioned above, the $f_{N}(x,y)\sim N(0,\sigma_{n}^{2})$ at (x,y). Hence, the noise level function becomes:(5)$$\sigma_{\eta }^{2}(x,y)=\rho \bar{f_{S}(x,y)}+\sigma_{n}^{2},$$
where $\sigma_{\eta }^{2}(x,y)$ is the local total noise variance of mixed noise model, $\bar{f_{S}(x,y)}$ is the expected intensity of noise-free image which is approximately equal to the local mean value $\bar{f(x,y)}$ of original noise image, ρ and $\sigma_{n}^{2}$ represent the Poisson noise parameter and Gaussian noise variance, respectively. In the PGN model, our mission was mainly to estimate the two parameters ρ and $\sigma_{n}^{2}$. We can calculate the local mean $\bar{f(x,y)}$ of the selected image blocks, and estimate the local total variance $\sigma_{\eta }^{2}(x,y)$. Then, a set of local mean-variance was extracted to accurately estimate ρ and $\sigma_{n}^{2}$ by MLE.

## 3. Proposed Noise Estimation Algorithm

### 3.1. Proposed Homogenous Blocks Selection Method

Primarily, a group of image blocks with size (M+1)×(N+1) from input noisy image of size R×C is generated by sliding the rectangular window pixel-by-pixel, and the group of blocks is expressed:(6)B=[b1, b2, …, bBN] ∈ℝR×C,
where bi∈ℝBN is the matrix of the *i*-th block, BN=(R−M)×(C−N) is the total number of blocks, and the local mean of nearly noise-free signal is computed via averaging all the noisy pixels in the block:(7)f(x,y)¯=1(M+1)(N+1)∑m=x−M/2;x+M/2∑n=y−N/2y+N/2f(m,n),
where the local window is around (x,y), and the window size is (M+1)×(N+1). Then, the proposed homogenous blocks selection method is introduced as following.

#### 3.1.1. Local Gray Statistic Entropy

In information theory, Shannon entropy [36] can effectively reflect the information contained in an event. The local gray statistic method is similar to Shannon entropy, but the new modified method focuses on how messy the gray texture values are. Firstly, given the original noisy image, the local map of normalized pixel grayscale value can be calculated as:(8)$$g(i,j)=\frac{f(i,j)+\xi }{\sum_{m=x-M/2;}^{x+M/2}\sum_{n=y-N/2}^{y+N/2}\left[f(m,n)+\xi \right]},$$
where i∈(x−M/2,x+M/2) and j∈(y−N/2,y+N/2) created the location (i,j) in local window around (x,y). Meanwhile f(i,j) is the gray level of original noisy image, g(i,j) is the normalized pixel grayscale value, and $\xi =10^{-9}$ is a small constant used to avoid the denominator becoming zero. Then, the proposed LGSE is defined as follows:(9)$$H_{le}(x,y)=-\left\{\sum_{i=x-1;}^{x+1}\sum_{j=y-1}^{y+1}g(i,j)\log[g(i,j)]+\sum_{i=x-M/2;}^{x+M/2}\sum_{j=y-N/2}^{y+N/2}g(i,j)\log[g(i,j)]\right\},$$
where $H_{le}(x,y)$ is the local gray entropy of 3 × 3 center pixels and all of the (M+1)×(N+1) neighbor pixels around the location (x,y). According to the theory of Shannon entropy, the LGSE reflects the degree of dispersion of local image grayscale texture. When the local image of gray distribution is uniform, the LGSE is relatively bigger, and when the local image of gray texture level distribution has a large dispersion, the LGSE is smaller. Due to the LGSE is the outcome of combined action of all pixels in the local window, the local entropy itself has a robust ability to resist noise. Six different noise-free textured blocks and their corresponding values of LGSE are given in Figure 2. As shown in Figure 2, the weakly textured blocks have relatively higher LGSE values. Note that here the weak texture is a generic term that includes not only smooth blocks, but also some slightly textured blocks. In Figure 3, the noisy image blocks are the blocks where the noise-free image blocks in Figure 2 are corrupted by AWGN at noise level $\sigma_{n}=20$, and their corresponding LGSE values are given. By comparing the results of Figure 2 and Figure 3, it is found that weakly textured blocks still have higher LGSE values even in the case of strong noise, while for the blocks with strong textures, their LGSE values drop faster. It can be seen from Figure 2 and Figure 3 that weakly textured blocks have bigger LGSE values and the proposed LGSE has a robust anti-noise ability for weakly textured blocks selection.

#### 3.1.2. Homogenous Blocks Selection Based on LGSE with Histogram Constraint Rule

Dong et al. [33] indicated that the image gray level histogram and block texture degree have a strong relationship. Inspired by this vision, to reduce or even overcome the disturbance of too many outliers, the gray level histogram-based constraint rule is introduced to firstly filter out the image blocks with disorderly rich textures.

In [19,28,30], the estimated noise level functions (NLFs) all have a comb shape in these scatterplots. These comb-shaped NLFs indicate that the pixel intensities approaching the highest or lowest pixel intensity may cause a disaster for noise estimation, because the brightest or darkest pixels may generate a number of outliers, ultimately reducing the precision of noise estimation. Therefore, we first exclude the blocks whose pixel mean value is too dark (0–15, the pixel intensity range is 0 to 255) or too bright (240–255) in the gray level histogram:(10)$$e_{i}=\{\begin{array}{cc} 0, & \bar{f_{i}}\geq 240\;or\;\bar{f_{i}}\leq 15\\ 1, & otherwise \end{array},$$
(11)$$b_{i}=e_{i}b_{i},$$
where $\bar{f_{i}}$ is the mean gray value of the *i*-th block, $e_{i}$ is the exclusion rule of darkest or brightest blocks, and $b_{i}$ is the updated values of *i*-th block by darkest or brightest block removal operation. $I_{r}\in [16,239]$ is the residual gray level of B after dark or bright block removal operation. The probability of occurrence of $I_{r}$ can be computed by:(12)$$p(I_{r})=\frac{hist(I_{r})}{T_{PN}},$$
where $hist(I_{r})$ denotes the frequency of $I_{r}$ in gray level histogram and $T_{PN}$ is the total pixel number of non-zero values of B. The blocks where gray values do not occur frequently may also have high LGSE values, which will generate some undesirable isolated values. Therefore, rather than dealing with all of gray levels, we only explore the gray levels with high occurrences. Accordingly, we try to eliminate the blocks with low frequency of occurrence below a certain threshold value. To solve this gap, the joint probability is firstly introduced as:(13)$$p[\phi (\bar{f_{i}})<thr,I_{r}]=\frac{\phi (\bar{f_{i}})<thr,\bar{f_{i}}=I_{r}}{T_{PN}},$$
where $\phi (\bar{f_{i}})=hist(\bar{f_{i}})/\max[hist(I_{r})]$ is the frequency of $\bar{f_{i}}$ versus maximum frequency of $I_{r}$ and thr is the threshold to evaluate and remove the blocks with low occurrence frequency of mean gray value. In terms of Bayes rule, the conditional probability can be expressed as:(14)$$\begin{array}{ll} p[\phi (\bar{f_{i}})<thr|I_{r}] & =\frac{p[\phi (\bar{f_{i}})<thr,I_{r}]}{p(I_{r})}\\ & =\frac{\phi (\bar{f_{i}})<thr,\bar{f_{i}}=I_{r}}{hist(I_{r})} \end{array}$$
By using the Equations (12–14), the blocks whose occurrence probability of their mean gray value $\bar{f_{i}}$ is smaller than the conditional probability value will be eliminated.

These procedures of gray level histogram-based constraint rule can be simply considered that we arrange the gray levels of pixels in a descending order, select the blocks of mean gray value $\bar{f_{i}}$ greater than *thr*, and renew block set B. To be specific, we set $thr=median(\phi (\bar{f_{i}}))$ to remove disorderly blocks. Hence, the blocks where the mean gray values do not occur frequently represent disorderly rich textured blocks, and the LGSE values of those blocks can be set as zero values. In Figure 4, the whole procedure of histogram-based constraint rule for eliminating outliers is shown using the “Birds” image as example. Figure 4a shows the original “Birds” image and its corresponding gray level histogram is shown in Figure 4b,c is the result of the selected gray levels using histogram constraint rule to overcome outliers. In Figure 4d, the green dots have covered the residual pixel blocks after the operation of histogram constraint rule.

The histogram constraint rule is a simple pretreatment to remove undesirable outliers but cannot obtain the reliable weakly textured blocks. To further suppress the disturbance of the textures of residual blocks, the LGSE-based homogenous image blocks selection method is proposed:(15)$$p[\tau \leq H_{le}(x,y)\leq \max H_{le}]=\delta ,$$
where τ is the selected minimum value of LGSE, $\max H_{le}$ is the maximum value of LGSE, and δ is the select ratio to control the weakly textured blocks selection. Based on the statistics and observation of noise estimation processing of 134 scene images, the select ratio is empirically set δ=10% of the total pixel number of whole images in our experiments. This setting means that we only selected 10% of the block set as homogenous blocks. An example for the homogenous blocks selection in “Birds” image is shown in Figure 5 and the 15 × 15 red boxes have outlined the finally selected homogenous blocks. The whole operations of homogenous blocks selection of “Barbara” and “House” are shown in Figure 6.

### 3.2. Haar Wavelet-Based Local Median Absolute Deviation for Local Variance Estimation

In this paper, a robust Haar wavelet-based local median absolute deviation (HLMAD) method is presented to estimate the standard deviation of noisy image block. The Haar wavelet, which is one of the simplest but most commonly used among wavelet transforms, can be exploited to detect noise for its good regularity and orthogonality. In the first scale of 2-D Haar wavelet analysis, the HH coefficients represent the high frequency component that is the noise component in selected homogenous blocks. The sub-band wavelet coefficients (HH) of first scale of 2-D Haar wavelet transform are calculated by:(16)$$\left\{ \begin{array}{l} H_{1x}(x,y)=\frac{f(x,y)-f(x+1,y)}{\sqrt{2}}\\ H_{2xy}(x,y)=\frac{H_{1x}(x,y)-H_{1x}(x,y+1)}{\sqrt{2}} \end{array}\right.,$$
where f(x,y) is the original noisy image, $H_{1x}(x,y)$ is the high frequency component of first step which uses 1-D Haar analysis in each row of the noisy image, and $H_{2xy}(x,y)$ is the high frequency coefficients of 2-D Haar wavelet analysis. Since the MAD is a more stable statistical deviation measure than sample variance, it works much better for more noise distributions. In addition, the MAD is more robust to outliers in datasets than sample standard deviation. By applying the robust MAD depicted in [15,16] to estimate the noise variance, the local block standard deviation ${\hat{\sigma }}_{\eta }(x,y)$ is derived as:(17)$$MAD[H_{2xy}(i,j)]=median(|H_{2xy}(i,j)|),$$
(18)$${\hat{\sigma }}_{\eta }(x,y)=\frac{MAD[H_{2xy}(i,j)]}{k},$$
where $MAD[H_{2xy}(i,j)]$ is the median value of absolute deviation of the high frequency coefficients of 2-D Haar wavelet $H_{2xy}(i,j)$, the constant of proportionality factor $k=\Phi^{-1}(3/4)=0.6745$, and ${\hat{\sigma }}_{\eta }(x,y)$ is the estimated standard deviation of the selected homogenous image block.

### 3.3. Maximum Likelihood Estimation for Multi-Parameter Estimation

According to the central limit theorem, the distribution of the sample approximately follows normal distribution when the number of sampling points is sufficient. Therefore, by fitting the NLF model $\sigma_{\eta }^{2}(x,y)=\rho \bar{f_{S}(x,y)}+\sigma_{n}^{2}$, the likelihood function of the dataset of selected homogenous blocks would be written as:(19)$$l(\rho ,\sigma_{n}^{2})=p(D|\rho ,\sigma_{n}^{2})=\prod_{i=1}^{T}p(d_{i}|\rho ,\sigma_{n}^{2})=\prod_{i=1}^{T}\frac{1}{\sqrt{2\pi \sigma_{\eta }^{2}(\bar{f_{i}};\rho ,\sigma_{n}^{2})}}\exp\left[\frac{-{\hat{\sigma }}_{\eta i}^{2}}{2\sigma_{\eta }^{2}(\bar{f_{i}};\rho ,\sigma_{n}^{2})}\right],$$
where T is the total number of selected blocks, dataset D is a set of local pixel mean value and local total variance pair $\left(\bar{f_{S}},{\hat{\sigma }}_{\eta }^{2}\right)$, and $d_{i}=(\bar{f_{i}},{\hat{\sigma }}_{\eta i}^{2})$, in here, $\bar{f_{i}}$ and ${\hat{\sigma }}_{\eta i}^{2}$ are the pixel mean value and the estimated noise variance of the *i*-th block. In addition, the ln-likelihood function is:(20)$$L(\rho ,\sigma_{n}^{2})=\ln[l(\rho ,\sigma_{n}^{2})]=-\frac{1}{2}\sum_{i=1}^{T}\left\{\ln[\sigma_{\eta }^{2}(\bar{f_{i}};\rho ,\sigma_{n}^{2})]+\frac{{\hat{\sigma }}_{\eta i}^{2}}{\sigma_{\eta }^{2}(\bar{f_{i}};\rho ,\sigma_{n}^{2})}+\ln(2\pi )\right\}.$$

According to the nature of maximum likelihood estimator, the two parameters $\hat{\rho }$ and ${\hat{\sigma }}_{n}^{2}$ could be estimated through maximizing the probability $L(\rho ,\sigma_{n}^{2})$ of the selected samples:(21)$$(\hat{\rho },{\hat{\sigma }}_{n}^{2})=\arg\underset{\left(\rho ,\sigma_{n}^{2}\right)}{\max}L(\rho ,\sigma_{n}^{2}).$$

Here, the gradient descent method is applied to solve the maximum likelihood function:(22)$$\{\begin{array}{l} \frac{\partial L(\rho ,\sigma_{n}^{2})}{\partial \rho }=0\\ \frac{\partial L(\rho ,\sigma_{n}^{2})}{\partial \sigma_{n}^{2}}=0 \end{array}.$$

And the two parameters $\hat{\rho }$ and ${\hat{\sigma }}_{n}^{2}$ of NLF can be estimated reliably.

### 3.4. Proposed Noise Parameter Estimation Algorithm

#### 3.4.1. Proposed Noise Parameter Estimation Algorithm for the AWGN Model

In the most commonly used image sensor noise model that is AWGN model, the noise standard deviation is the only parameter that needs to be estimated. According to these above mentioned methods and theories, the accurate noise standard deviation estimation algorithm is established. Firstly, we divide the noisy image into a group of blocks by sliding window pixel-by-pixel, and compute the mean gray value of each block. The LGSE of each block is calculated, and the LGSE values of darkest and brightest blocks are excluded. Next, the LGSE values of residual blocks make a descending order, and the homogenous blocks can be extracted by selecting several largest LGSE values. Then, the local standard deviation of each selected block is computed by the HLMAD, and build up a cluster of local standard deviation. Finally, the noise standard deviation is precisely estimated by picking the median of the cluster:(23)$${\hat{\sigma }}_{n}=median({\hat{\sigma }}_{\eta }),$$
where ${\hat{\sigma }}_{\eta }$ is the cluster of local standard deviation ${\hat{\sigma }}_{\eta }(x,y)$, ${\hat{\sigma }}_{n}$ is the estimated noise standard deviation for AWGN model. The whole of the proposed noise parameter estimation algorithm for the CCD sensor AWGN model is summarized in Algorithm 1.

**Algorithm 1.** Proposed noise parameter estimation algorithm for the AWGN modelInput: Noisy image f, window size N, blocks selection ratios δ.Output: Estimated noise standard deviation σ^n.Step 1: Group a set of blocks by sliding window pixel-by-pixel:B=[b1, b2, …, bBN] ∈ℝR×C.Step 2: Compute the mean gray value f(x,y)¯ of all pixels at each block.Step 3: **for** block index k = 1:BN **do**Compute the LGSE of blocks according to (8) and (9).
**end for**
Step 4: Exclude the LGSE of blackest and whitest blocks according to (10) and (11).Step 5: Select homogenous blocks based on selection of residual LGSE using (15).Step 6: **for** homogenous block index t = 1:T **do**Obtain the local standard deviation of homogenous block by HLMAD according to (16–18).
**end for**
Step 7: Estimate the noise standard deviation ${\hat{\sigma }}_{n}^{}$ via using median estimator derived from (23).

#### 3.4.2. Proposed Noise Parameter Estimation Algorithm for the PGN Model

We also work on the PGN model, which is the most likely model for practical image sensor noise. It is noteworthy that the proposed estimation algorithm for the PGN model generally agrees with the proposed algorithm for AWGN model. Contrasted with Algorithm 1, for the PGN model, we append the gray level histogram-based constraint rule method for firstly removing disorderly blocks, and compute the local mean value of selected blocks. Then, the local mean-variance pair $\left(\bar{f},{\hat{\sigma }}_{\eta }^{2}\right)$ of selected blocks can be obtained. Finally, by utilizing the MLE estimator to process the local mean-variance pair $\left(\bar{f},{\hat{\sigma }}_{\eta }^{2}\right)$, the two parameters $\hat{\rho }$ and ${\hat{\sigma }}_{n}^{2})$ of PGN model are estimated credibly. The whole of the proposed noise parameter estimation algorithm for the image sensor PGN model is summarized in Algorithm 2.

**Algorithm 2.** Proposed noise parameter estimation algorithm for the PGN modelInput: Noisy image f, window size N, blocks selection ratios δ.Output: Estimated NLF parameters ρ^ and σ^n2.Step 1: Group a set of blocks by sliding window pixel-by-pixel:B=[b1, b2, …, bBN] ∈ℝR×C.Step 2: Compute the mean gray value f(x,y)¯ of all pixels in each block.Step 3: **for** block index k = 1:BN **do**Compute the LGSE of blocks according to (8) and (9).
**end for**
Step 4: Exclude the LGSE of blackest and whitest blocks according to (10) and (11).Step 5: Utilize the gray level histogram-based constraint rule to remove disorderly blocks,and exclude the LGSE of low frequency gray value according to (12–14).Step 6: Select homogenous blocks based on selection of residual LGSE using (15).Step 7: **for** homogenous block index t = 1:T **do**Compute the mean gray value of all pixels in selected block using (7).Obtain the local variance of homogenous block by HLMAD according to (16–18).
**end for**
Step 8: Estimate the two parameters $\hat{\rho }$ and ${\hat{\sigma }}_{n}^{2}$ of NLF via processing local mean-variance pair $\left(\bar{f},{\hat{\sigma }}_{\eta }^{2}\right)$ using MLE derived from (19)–(22).

## 4. Experimental Results

In this section, a series of experiments on synthesized noisy images under various scene images are conducted to evaluate the performance of proposed noise estimation algorithm. Furthermore, some state-of-the-art algorithms and classical algorithms are selected for performance comparison. These experiments are performed in Matlab 2016a (developed by MathWorks, Massachusetts, USA) on the computer with 3.2 Ghz Intel i5-6500 CPU and 16 Gb random access memory.

### 4.1. Test Dataset

The test dataset is composed of 134 images for synthesized noisy images, which were generated by adding the AWGN or PGN component to three typical test datasets. These three datasets consisted of classic standard test images (CSTI), Kodak PCD0992 images [37] and Berkeley segmentation dataset (BSD) images [38]. The CSTI images have 10 classic standard test images of size 512 × 512 and are shown in Figure 7. The Kodak PCD0992 images have 24 images of size 768 × 512 or 512 × 768 released by the Eastman Kodak Company for unrestricted usage, and many researchers use them as a standard test suite for image de-noising testing. The BSD images totally have 100 test images and 200 train images of size 481 × 321 or 321 × 481, but in our experiments, we randomly selected 25 images from the Berkeley test set and 75 images from Berkeley train set to process. Therefore, the test dataset contains various scene images. Testing on this dataset proves that the algorithm is suitable for many scenarios.

### 4.2. Results of Noise Estimation

#### 4.2.1. Effects of Parameters

In this part, the experiment is conducted to compare the performance of proposed noise estimation algorithm under various parameter configurations. To qualify the accuracy of the proposed noise estimation algorithm, the mean error of noise level estimation is used to compare the effects of different parameters:(24)$$V_{ME}=\frac{1}{31}\sum_{n=0}^{30}{\left(\bar{\sigma_{est}^{2}(n)}-\sigma_{added}^{2}(n)\right)}^{2},$$
where $\sigma_{added}^{2}$ is the added noise variance, $\sigma_{est}^{2}$ is the group of estimated noise variances, and $\bar{\sigma_{est}^{2}}$ is the average value of estimated noise variance. In this parameter selection experiment, we set noise level n∈[0,30]. Obviously, the more accurate the noise estimation algorithm is, the smaller the mean error is.

The proposed noise estimation algorithm is a block-based method, and the performance of proposed method is related to block size. In this experiment, the test dataset with selected 134 images was processed by the proposed noise estimation algorithm with different block sizes. The mean error-bars of different block sizes are shown in Figure 8, and it indicates that the block size N=15 can be more suitable for these noisy scene images.

#### 4.2.2. Comparison to AWGN Estimation Baseline Methods

In this part, four classical noise level estimation methods and three state-of-the-art noise level estimation methods are introduced in the comparison experiments to evaluate the performance of the proposed AWGN level estimation method. The Immerkær’s fast noise variance estimation (FNVE) [9], Khalil’s median absolute deviation (MAD) based noise estimation [39], Santiago’s variance mode (VarMode) noise level estimation [40] and Zoran’s discrete cosine transform (DCT) based noise estimation [12] were chosen as the classical noise level estimation methods. The Olivier’s nonlinear noise estimator (NOLSE) [11], Pyatykh’s principal component analysis (PPCA) based noise esti-mation [17] and Lyu’s noise variance estimation (EstV) [13] were selected as the state-of-the-art noise estimation methods.

Figure 9 shows three example images (“Barbara”, “House”, “Birds”) which were selected from three image datasets, and the corresponding noise level estimation results were obtained by the compared noise estimation methods. The original images are shown in the first row. Figure 9d–f shows their results of noise level estimation respectively. As shown in Figure 9, the DCT-based noise estimator performs better at low noise level, but underestimates the noise at higher noise level. FNVE, MAD, VarMode and NOLSE noise estimators easily overestimate the noise level. EstV noise estimator underestimates the noise level in each instance. For most noise levels, both the PPCA-based AWGN estimator and the proposed AWGN estimator work well. In these three scenes, our proposed method outperforms these several existing noise estimation methods.

To further prove the capability of the proposed method, we expanded our experiments on test dataset with 134 images. Table 1, Table 2 and Table 3 show the results of different AWGN estimation algorithms on the CSTI images, the Kodak PCD0992 images and the BSD images, respectively. In Table 1, Table 2 and Table 3, we set the added noise levels $\sigma_{n}$ = 0, 1, 5, 10, 15, 20 and 25. As shown in the Table 1, Table 2 and Table 3, FNVE, MAD, NOLSE and VarMode estimators overestimate the noise level. This is because these filter-based methods fail to fully consider the effects of image textures on noise estimation. DCT and EstV estimators work well for low noise levels, even better than the proposed estimator when the noise level is less than five, but seriously underestimate the noise level at high noise conditions. The reason is that these two methods use band-pass statistical kurtosis to extract noise components and eliminate the interference of textures to noise estimation. For higher noise levels, these estimators remove too many noise components while removing textures, which leads to underestimation of noise. Compared with those above methods, the PPCA estimator has a better performance. In most cases, the proposed AWGN estimation method is superior to the PPCA estimator. Generally speaking, the experimental results prove that the proposed AWGN estimator not only is suitable for most noise levels, but also has the better estimation capability compared with those state-of-the-art methods.

#### 4.2.3. Comparison to PGN Estimation Baseline Methods

In this part, the comparison with four representative PGN estimation methods is conducted to evaluate the proposed PGN estimation method. The Foi’s clipped Poisson-Gaussian (CPG) noise fitting algorithm [28], Zabrodina’s scatter-plots regression curve fitting (RCF) estimator [29], Jeong’s simplified Poisson-Gaussian noise (SPGN) estimator [41] and Liu’s principal component analysis (LPCA) based noise estimator [35] were selected as baseline methods because they are well studied and widely used for assessing new PGN estimator.

Figure 10 shows three example images, and their corresponding estimated NLF results. The noisy images with ρ = 0.5 and $\sigma_{n}^{2}$ = 10 are shown in the first row. The second row displays their NLF results of the proposed estimation method respectively. The third row shows the estimated NLF results of different PGN estimation methods respectively. It can be seen in Figure 10 that, the CPG, SPGN and RCF noise estimation methods overestimate the NLF in three examples, while both the LPCA-based PGN estimator and proposed PGN estimator work well, and the proposed method works better than LPCA on “Barbara” and “Birds” at noise parameters ρ = 0.5 and $\sigma_{n}^{2}$ = 10.

To verify the performance of the proposed PGN estimation method, we still insisted on performing experiments on the test dataset with 134 images. The performances of the compared PGN estimation methods for different image datasets are shown in Table 4, Table 5 and Table 6. In Table 4, Table 5 and Table 6, we set the added PGN parameters ($\rho ,\sigma_{n}^{2}$) = (0.1,1), (0.1,5), (0.1,10), (0.5,1), (0.5,5), (0.5,10). As shown in the Table 4, Table 5 and Table 6, CPG, RCF and SPGN estimators overestimate the parameters of NLF. Table 4 and Table 5 depict that the proposed PGN estimator work better than other PGN estimators on CSTI images and Kodak PCD0992 images in all cases. Table 6 shows that both the proposed PGN estimator and LPCA-based PGN estimator have comparable estimation results and are better than other estimators on BSD images. This indicates that the LPCA-based PGN estimator performs well for BSD images of size 481×321 but the proposed PGN estimator can obtain the best performance for all three image datasets, which means that the proposed PGN estimator can work more stably for images with different sizes. Consequently, compared with these representative PGN estimators, the proposed PGN estimator shows more accuracy for various scenarios at different PGN parameters.

### 4.3. Noise Estimation Tuned for Blind De-Noising

#### 4.3.1. Quantify the Performance of Blind De-Noising

Blind de-noising is a pre-processing application that needs the estimated noise parameters. In this part, we try to prove that the proposed AWGN estimator and proposed PGN estimator can promote the performance of existing blind de-noising algorithms. In this part, we chose the state-of-the-art de-noising algorithm BM3D [42] to remove the estimated AWGN of noisy image, and selected representative PGN de-noising algorithm VST-BM3D [43] to eliminate the estimated PGN of noisy image.

In order to make a reliable comparison of similarity between noise-free image and de-noised image, two image quality assessment methods were used in this paper: (1) The structural similarity index measurement (SSIM) [44] and (2) the peak signal-to-noise ratio (PSNR). The SSIM is defined as:(25)$$SSIM(f,f_{S})=\frac{\left(2u_{f}u_{f_{S}}+C_{1})(2\sigma_{f,f_{S}}+C_{2}\right)}{\left(u_{f}^{2}+u_{f_{S}}^{2}+C_{1})(\sigma_{f}^{2}+\sigma_{f_{S}}^{2}+C_{2}\right)},$$
where f is the noisy image, $f_{S}$ is the noise-free image, $u_{f}$ and $u_{f_{S}}$ are the mean values of noisy image and noise-free image, respectively; $\sigma_{f}$, $\sigma_{f_{S}}$ and $\sigma_{f,f_{S}}$ are standard deviations of noisy image, noise-free image and their covariance, respectively; $C_{1}$ and $C_{2}$ are small positive constants used to avoid the denominator becoming zero. The PSNR is written as:(26)$$PSNR=10\log(\frac{255^{2}}{MSE}),$$
where MSE is the mean squared error of noisy image and can be derived from:(27)$$MSE=\frac{1}{RC}\sum_{r=0}^{R-1}\sum_{c=0}^{C-1}{\left(||f(r,c)-f_{S}(r,c)||\right)}^{2},$$
where the size of input noisy image is R×C, and (r,c) is the pixel location.

#### 4.3.2. Noise Estimation Tuned for AWGN Blind De-Noising

To quantitatively evaluate the performance of proposed noise estimation algorithm for blind de-noising, all 134 test images of the test dataset were used in AWGN blind de-noising experiment. The added noise levels were set as $\sigma_{n}$ = 1, 5, 10, 15, 20 and 25 in this experiment. The average SSIM and PSNR comparisons of different AWGN estimators with BM3D for blind de-noising on the test dataset are shown in Figure 11. Figure 11a depicts the SSIM values of different AWGN estimators with BM3D for blind de-noising, and the proposed estimator with BM3D de-noising algorithm has the biggest SSIM values at different noise levels. This demonstrates that the proposed AWGN estimator can promote the structure similarity index of blind de-noising. Figure 11b shows the proposed estimator with BM3D de-noising algorithm has the highest bars of PSNR value at different noise levels. This indicates that the proposed AWGN estimator can decrease the distortion of blind de-noising. In summary, the proposed AWGN estimator can increase the performance of blind de-noising stably.

#### 4.3.3. Noise Estimation Tuned for PGN Blind De-Noising

Figure 12 shows the average SSIM and PSNR values of different PGN estimators with VST-BM3D for blind de-noising at different noise parameters, and the proposed PGN estimator with VST-BM3D de-noising algorithm has the largest SSIM and PSNR values at different noise parameters and is superior to LPCA. This demonstrates that the proposed PGN estimator can improve the performance of PGN blind de-noising. From above experiments, it can be seen that the proposed AWGN and PGN estimators not only can work stably for various scene images with different noise parameters, but also can promote the performance of blind de-noising.

## 5. Conclusions and Future Work

A novel method based on the properties of LGSE and HLMAD is presented in this paper to estimate the AWGN and PGN parameters of an image sensor at different noise levels. By using the map of LGSE, the proposed method can select homogenous blocks from an image with complex textures and strong noise. The local noise variances of the selected homogenous blocks are efficiently computed by the HLMAD, and the noise parameters can be estimated accurately by the MLE. Extensive experiments verify that the proposed AWGN and PGN estimation algorithms have a better estimation performance than compared state-of-the-art methods, including FNVE, MAD, VarMode, DCT, NOLSE, PPCA, EstV for the AWGN model and CPG, RCF, SPGN, LPCA for the PGN model. The experimental results also demonstrate that the proposed method can work stably for various scene images at different noise levels and can improve the performance of blind de-noising. Therefore, combing the proposed noise estimators with recent de-noising methods to develop a novel blind de-noising algorithm, which can remove image sensor noise and preserve fine details effectively for actual noisy scene images, is one important direction in our future studies.

## Figures and Tables

**Figure 1 sensors-19-00339-f001:**
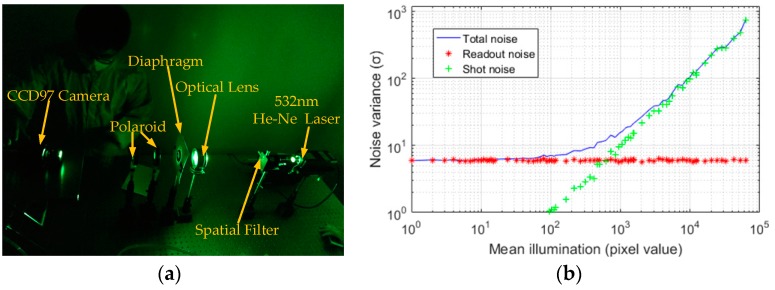
Real noise analysis for charge-coupled device (CCD) image sensor. (**a**) Experimental platform for noise analysis of CCD image sensor; (**b**) computed the Photon Transfer Curve (PTC) according to the measurement of [5,6].

**Figure 2 sensors-19-00339-f002:**
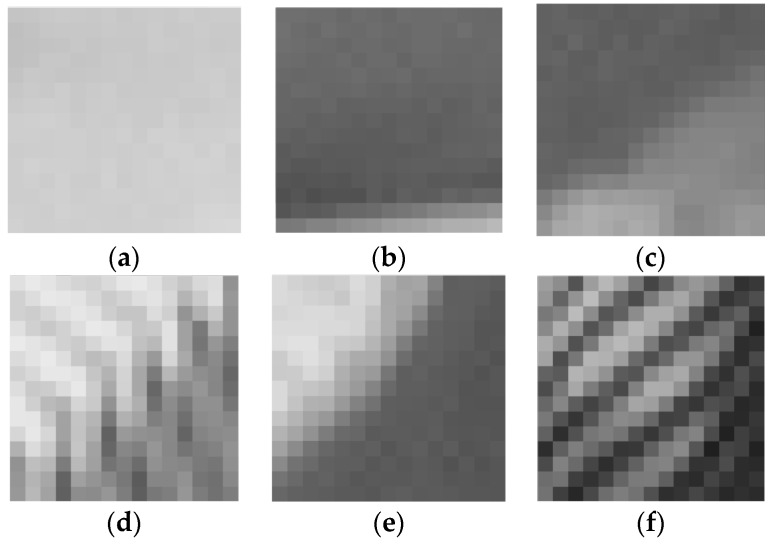
Local gray statistic entropy $H_{le}$ of various noise-free image blocks, and weakly textured blocks have relatively bigger local gray statistic entropy (LGSE) values; (**a**) $H_{le}$ = 7.8135; (**b**) $H_{le}$ = 7.7972; (**c**) $H_{le}$ = 7.7761; (**d**) $H_{le}$ = 7.7618; (**e**) $H_{le}$ = 7.7158; (**f**) $H_{le}$ = 7.6994.

**Figure 3 sensors-19-00339-f003:**
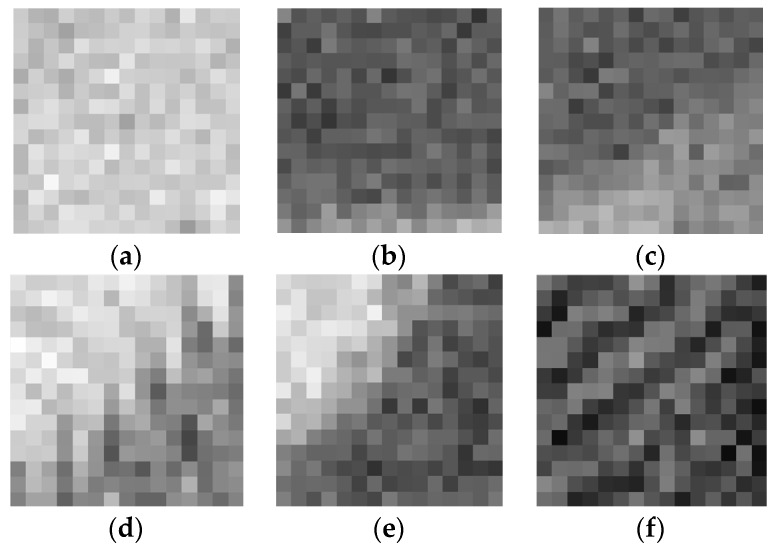
Local gray statistic entropy $H_{le}$ of noisy image blocks corrupted by additive white Gaussian noise (AWGN) at noise level $\sigma_{n}=20$, and weakly textured blocks have bigger LGSE values; (**a**) $H_{le}$ = 7.8094; (**b**) $H_{le}$ = 7.7155; (**c**) $H_{le}$ = 7.6537; (**d**) $H_{le}$ = 7.4976; (**e**) $H_{le}$ = 7.3127; (**f**) $H_{le}$ = 7.1362.

**Figure 4 sensors-19-00339-f004:**
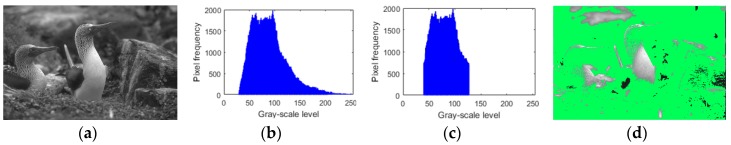
An example for histogram-based constraint rule for eliminating outliers. (**a**) Original “Birds” image; (**b**) the gray level histogram of (**a**); (**c**) the selected gray levels after the operation of histogram constraint rule; (**d**) The selected blocks according to (**c**) are labeled with green color.

**Figure 5 sensors-19-00339-f005:**
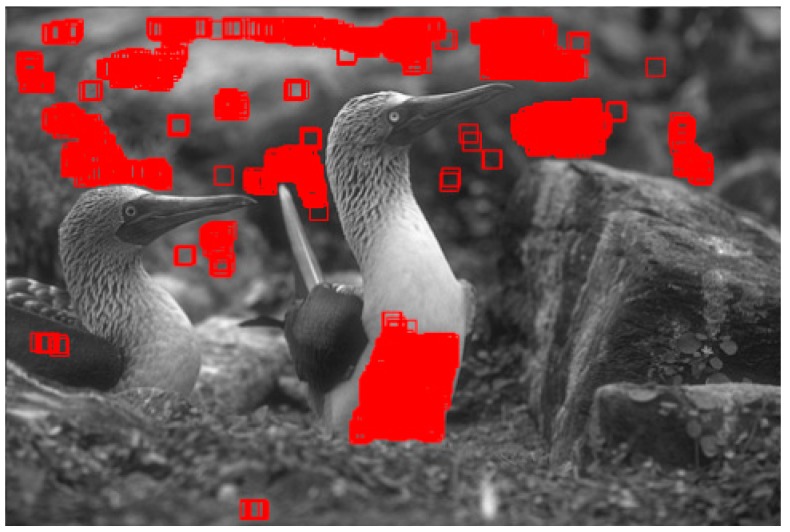
An example for homogenous blocks selection, and those 15 × 15 red boxes in the “Birds” image have outlined the finally selected homogenous blocks.

**Figure 6 sensors-19-00339-f006:**
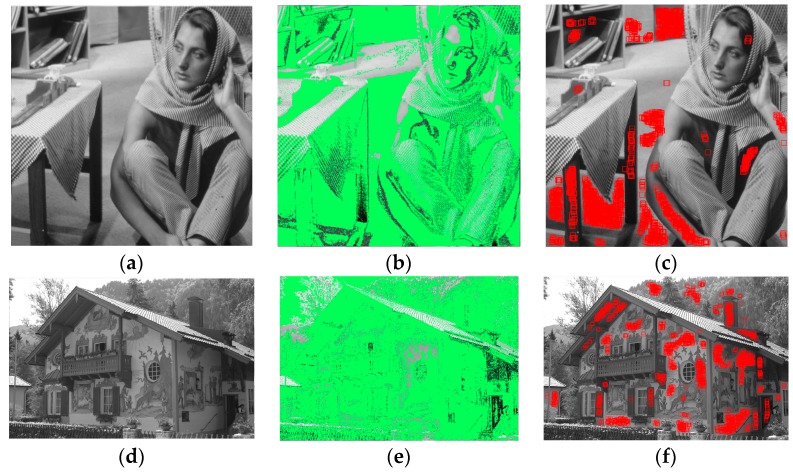
Two examples for homogenous blocks selection in the “Barbara” and “House.” (**a**) Original “Barbara” image; (**b**) the selected blocks after histogram constraint in “Barbara” image are labeled with green color; (**c**) 15 × 15 red boxes in the “Barbara” image have outlined the finally selected homogenous blocks; (**d**) original “House” image; (**e**) the selected blocks after histogram constraint in the “House” image are labeled with green color; (**f**) 15 × 15 red boxes in the “House” image have outlined the finally selected homogenous blocks.

**Figure 7 sensors-19-00339-f007:**
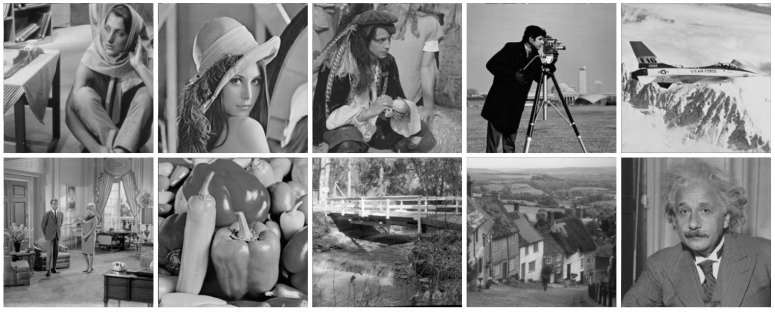
Classic standard test images for synthesized noisy images: “Barbara,” “Lena,” “Pirate,” “Cameraman,” “Warcraft,” “Couple,” “Peppers,” “Bridge,” “Hill” and “Einstein.”

**Figure 8 sensors-19-00339-f008:**
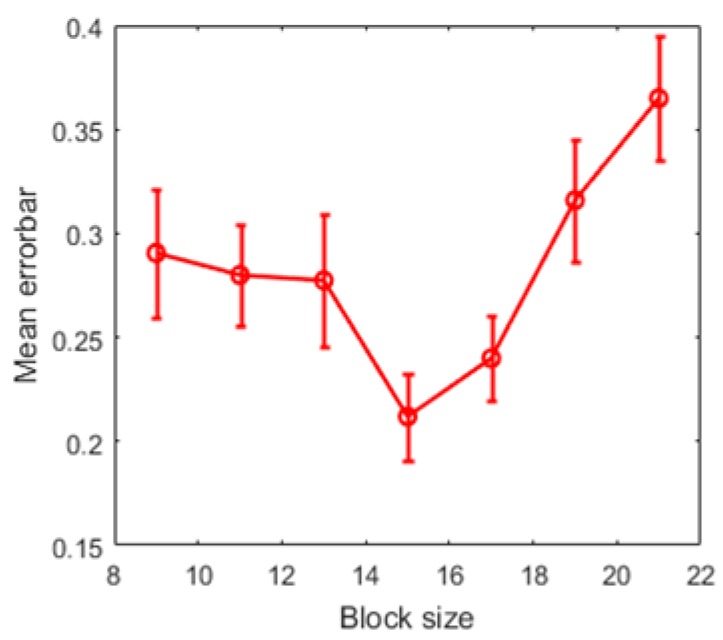
The mean error-bars of different block sizes. Thus, when the block size N=15, the proposed algorithm has an overwhelming advantage for its accuracy and stability.

**Figure 9 sensors-19-00339-f009:**
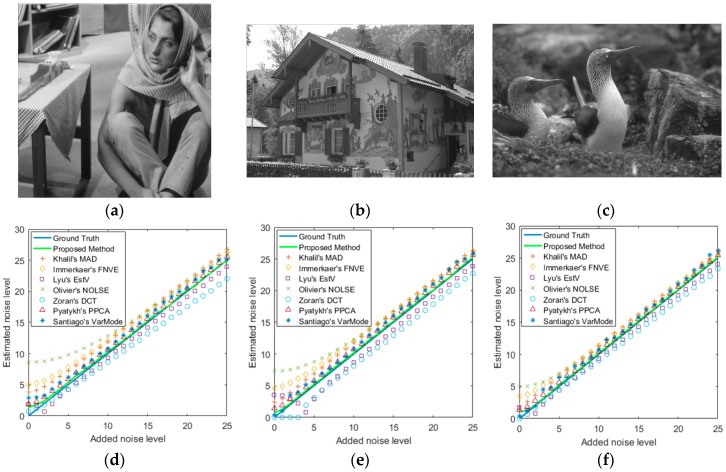
Different AWGN level estimation methods on three images, and the proposed method outperforms other several existing noise estimation methods. (**a**–**c**) are original images (“Barbara,” “House,” “Birds”); (**d**–**f**) are their results of noise level estimation respectively.

**Figure 10 sensors-19-00339-f010:**
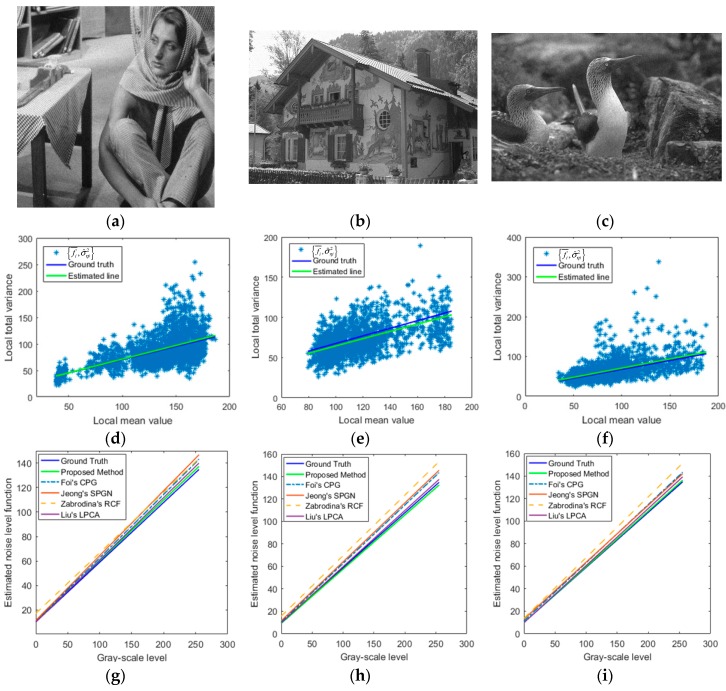
Different Poisson-Gaussian noise (PGN) level estimation methods on three images, and the proposed method out-performs several existing PGN estimation methods. (**a**–**c**) are noisy images (“Barbara,” “House,” “Birds”) with ρ=0.5 and $\sigma_{n}^{2}=10$; (**d**–**f**) are the results of our parameter estimation algorithm respectively; (**g**–**i**) are the results of different PGN estimation methods respectively.

**Figure 11 sensors-19-00339-f011:**
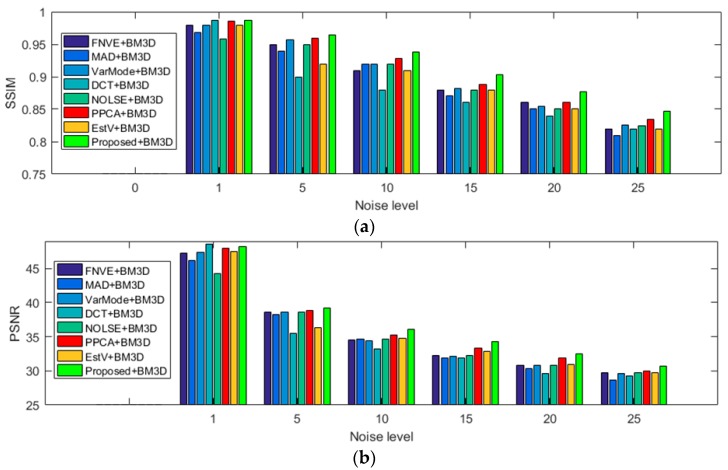
Structural similarity index measurement (SSIM) and peak signal-to-noise ratio (PSNR) comparison results of different AWGN estimators with BM3D for blind de-noising on the 134 selected images at different noise levels. (**a**) Comparison results of SSIM; (**b**) comparison results of PSNR.

**Figure 12 sensors-19-00339-f012:**
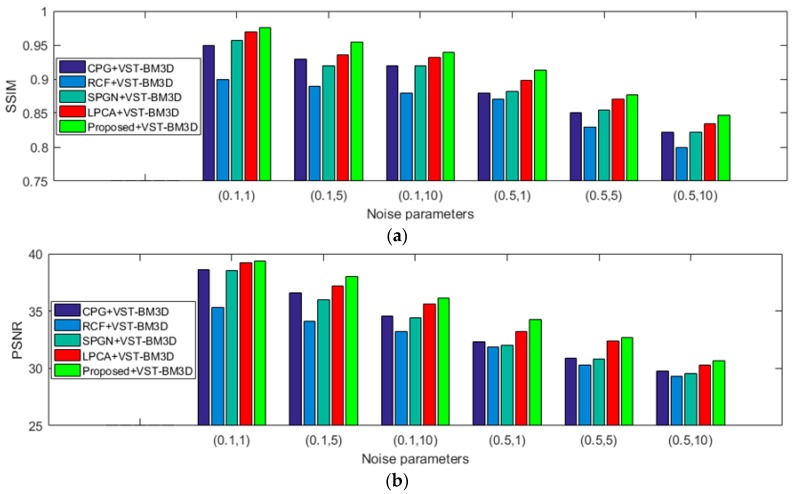
SSIM and PSNR comparison results of different PGN estimators with VST-BM3D for blind de-noising on the 134 selected images at different noise parameters. (**a**) Comparison results of SSIM; (**b**) comparison results of PSNR.

**Table 1 sensors-19-00339-t001:** The results of different AWGN estimation algorithms on the classic standard test images (CSTI).

Ground Truth	Immerkær’s FNVE	Khalil’s MAD	Santiago’s VarMode	Zoran’s DCT	Olivier’s NOLSE	Pyatykh’s PPCA	Lyu’s EstV	Our Proposed
$\sigma_{n}=0$	3.31	3.03	2.52	1.41	3.92	**1.29**	1.39	1.77
$\sigma_{n}=1$	3.57	3.25	2.91	1.70	4.05	1.72	**1.65**	1.91
$\sigma_{n}=5$	6.40	6.34	6.23	**5.02**	6.406	5.48	5.10	5.34
$\sigma_{n}=10$	10.92	10.99	10.94	9.64	10.70	10.31	**9.87**	**10.13**
$\sigma_{n}=15$	15.72	15.80	15.81	14.39	15.42	15.25	14.74	**15.11**
$\sigma_{n}=20$	20.59	20.67	20.56	19.21	20.27	20.13	19.57	**19.88**
$\sigma_{n}=25$	25.57	25.60	25.58	24.12	25.22	25.11	24.42	**25.01**

**Table 2 sensors-19-00339-t002:** The results of different AWGN estimation algorithms on the Kodak PCD0992 images.

Ground Truth	Immerkær’s FNVE	Khalil’s MAD	Santiago’s VarMode	Zoran’s DCT	Olivier’s NOLSE	Pyatykh’s PPCA	Lyu’s EstV	Our Proposed
$\sigma_{n}=0$	3.43	3.16	2.67	1.32	3.96	1.32	1.48	**1.18**
$\sigma_{n}=1$	3.58	3.13	2.50	**1.34**	4.53	1.45	1.43	1.75
$\sigma_{n}=5$	6.47	6.49	5.75	4.53	6.67	5.21	4.39	**5.14**
$\sigma_{n}=10$	11.01	11.18	10.69	9.51	10.88	**10.08**	9.57	10.11
$\sigma_{n}=15$	15.79	16.00	15.61	14.37	15.56	15.15	14.56	**15.10**
$\sigma_{n}=20$	20.67	20.85	20.64	19.23	20.38	19.87	19.51	**20.04**
$\sigma_{n}=25$	25.61	25.74	25.59	24.12	25.30	24.89	24.44	**25.00**

**Table 3 sensors-19-00339-t003:** The results of different AWGN estimation algorithms on the Berkeley segmentation dataset (BSD) images.

Ground Truth	Immerkær’s FNVE	Khalil’s MAD	Santiago’s VarMode	Zoran’s DCT	Olivier’s NOLSE	Pyatykh’s PPCA	Lyu’s EstV	Our Proposed
$\sigma_{n}=0$	3.12	2.37	1.56	**0.58**	4.40	0.78	1.75	0.66
$\sigma_{n}=1$	3.49	2.86	2.05	**0.94**	4.49	1.32	1.77	1.37
$\sigma_{n}=5$	6.51	6.49	5.86	4.55	6.62	5.31	4.49	**5.19**
$\sigma_{n}=10$	11.06	11.21	10.63	9.37	10.83	10.29	9.47	**10.17**
$\sigma_{n}=15$	15.85	16.03	15.75	14.26	15.50	15.18	14.42	**15.05**
$\sigma_{n}=20$	20.74	20.85	20.54	19.19	20.34	20.15	19.33	**19.98**
$\sigma_{n}=25$	25.70	25.77	25.66	24.18	25.26	25.09	24.21	**24.96**

**Table 4 sensors-19-00339-t004:** The results of different PGN estimation algorithms on the CSTI images.

Ground Truth	Foi’s CPG	Zabrodina’s RCF	Jeong’s SPGN	Liu’s LPCA	Our Proposed
ρ=0.1	0.136	0.124	0.126	**0.108**	0.091
$\sigma_{n}^{2}=1$	0.02	2.14	0.35	0.03	**1.18**
ρ=0.1	0.131	0.135	0.130	0.052	**0.088**
$\sigma_{n}^{2}=5$	5.36	7.73	5.42	5.98	**5.32**
ρ=0.1	0.142	0.159	0.136	0.064	**0.073**
$\sigma_{n}^{2}=10$	10.61	14.52	9.81	10.62	**10.16**
ρ=0.5	0.518	0.612	0.607	0.456	**0.481**
$\sigma_{n}^{2}=1$	1.25	2.67	1.63	1.35	**1.24**
ρ=0.5	**0.521**	0.618	0.587	**0.479**	0.462
$\sigma_{n}^{2}=5$	5.44	7.09	6.00	**5.18**	5.21
ρ=0.5	0.522	0.601	0.561	0.525	**0.517**
$\sigma_{n}^{2}=10$	10.86	13.65	10.53	8.97	**9.61**

**Table 5 sensors-19-00339-t005:** The results of different PGN estimation algorithms on the Kodak PCD0992 images.

Ground Truth	Foi’s CPG	Zabrodina’s RCF	Jeong’s SPGN	Liu’s LPCA	Our Proposed
ρ=0.1	0.157	0.143	0.146	0.061	**0.076**
$\sigma_{n}^{2}=1$	0.01	1.72	1.57	1.50	**1.35**
ρ=0.1	0.145	0.183	0.158	0.123	**0.081**
$\sigma_{n}^{2}=5$	5.63	6.22	**4.69**	5.65	5.66
ρ=0.1	0.142	0.182	0.166	**0.121**	**0.079**
$\sigma_{n}^{2}=10$	10.50	10.54	10.89	9.33	**10.06**
ρ=0.5	0.538	0.661	0.596	0.470	**0.479**
$\sigma_{n}^{2}=1$	**1.23**	1.63	1.31	1.26	1.27
ρ=0.5	0.547	0.627	0.558	0.481	**0.482**
$\sigma_{n}^{2}=5$	5.25	5.98	5.71	5.34	**5.13**
ρ=0.5	0.534	0.607	0.572	0.533	**0.526**
$\sigma_{n}^{2}=10$	10.91	10.86	9.22	9.16	**9.54**

**Table 6 sensors-19-00339-t006:** The results of different PGN estimation algorithms on the BSD images.

Ground Truth	Foi’s CPG	Zabrodina’s RCF	Jeong’s SPGN	Liu’s LPCA	Our Proposed
ρ=0.1	0.127	0.138	0.136	**0.092**	0.086
$\sigma_{n}^{2}=1$	0.20	1.50	0.27	0.08	**1.39**
ρ=0.1	0.137	0.146	0.143	0.082	**0.083**
$\sigma_{n}^{2}=5$	**4.63**	6.01	5.46	6.03	5.95
ρ=0.1	0.139	0.167	0.157	**0.071**	0.065
$\sigma_{n}^{2}=10$	9.75	10.32	10.88	**10.16**	10.18
ρ=0.5	**0.514**	0.652	0.583	**0.486**	0.481
$\sigma_{n}^{2}=1$	1.13	1.83	1.42	1.13	**1.04**
ρ=0.5	0.516	0.639	0.565	0.472	**0.485**
$\sigma_{n}^{2}=5$	5.18	5.72	5.23	5.12	**5.11**
ρ=0.5	0.521	0.618	0.593	**0.509**	**0.491**
$\sigma_{n}^{2}=10$	**10.22**	10.49	8.64	9.62	9.73

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
