# Peer review of "Noise Estimation for Image Sensor Based on Local Entropy and Median Absolute Deviation"

_sensors, 2019, doi:10.3390/s19020339_

Reviewer 1 Report

This paper presents a method to estimate the two most common varieties of image sensor noise, Average White Gaussian Noise (AWGN), and Poisson Gaussian Noise (PGN).

The method presented first selects the subset of the image more adequate to estimate the target noise. In case of AWGN, homogeneous regions are selected, while, for PGN, weakly textured regions are selected.

Then, the noise parameters are estimated from the statistics of the regions selected.

The presented algorithms are compared to state-of-the-art alternatives and show competitive results on estimating noise parameters as well as using those estimations to perform blind de-noising.

Strengths of the paper:

The overall conception of the paper is clear, and well structured. Related work is well covered and described. The experimental section is methodical and extensive.

Weaknesses:

The presentation clarity is very inconsistent. Some parts are very well written, while others (i.e., captions), seem to have been put together in a  rush and are difficult to understand.

The method seems to be very much ad-hoc, and relies on several steps with magic values that are weakly justified. Hence it is unclear if the algorithm was hand tuned to top state-of-the-art algorithms. Furthermore, the algorithm descriptions are extremely imprecise, which sentences such as "Utilize the gray level histogram-based method to roughly select weakly textured blocks". A concept like "roughly select" is valuable to provide an insight of what the method is doing, but is totally inappropriate for an "algorithm" description which should be a concise, faithful and precise description of what is your method doing.

It would be appreciated if the section could be reorganized to improve legibility altogether.

Equations. Some of them are not very informative or directly redundant (i.e., 1-4,15, 24). Parenthesis size is not adapted to the number of inner parenthesis (i.e., 9, 16). Commas in equations that are followed by a capital letter, instead of a full stop (i.e., 14,15). Fonts seem not embedded and some symbols are not well rendered (i.e., 11). 

Reviewer 2 Report

Contributions:

This study presents a homogenous block-based noise estimation method for an image corrupted by additive white Gaussian noise and Poisson-Gaussian noise. The improvement is not significant. I think this paper requires major revisions. Some comments are given below:

Comments:

1.(Lines 202-206 of page 5)The authors stated “When the local image of gray distribution is relatively uniform, the LGSE is much bigger, and when the local image of gray level distribution in large dispersion, the LGSE is smaller. Due to the LGSE is the outcome of combined action of all pixels in the local window, the local entropy itself has a robust ability to resist noise. Six different textured blocks and their corresponding values of LGSE are given in Figure 2.”. By observing Fig. 2, the values of LGSE vary slightly for different textures. The statements “When the local image of gray distribution is relatively uniform, the LGSE is much bigger” are not adequate. Accordingly, I cannot believe that the LGSE is a robust feature.  

2.(Lines 391 to 393 of page 13) The authors stated “Consequently, the experimental results prove that the proposed AWGN estimator not only is suitable for most noise levels, but also has the better estimation capability compared with those state-of-the-art methods.”. By observing Tables 1-6, the proposed method cannot outperform some compared methods. However, the authors did not provide the reason. In addition, the improvement is very slight for the proposed method.

3.(Page 18) In page 10, please mark the improvement positions for the proposed method. The denoised images are very similar to the compared methods, in particular for the denoised images in Figs. 10(b),(c), (d), (f), and (g).

4.(Page 19) In page 11, please mark the improvement positions for the proposed method. The denoised images are very similar to the compared methods, in particular for the denoised images in Figs. 11(b), (d), and (e).

5.(Line 53 of page 2)”In reference [4],” should be revised as “”In [4],”. Line 86 of page

2 and line 219 of page 6 also have the same problem.

6.(Lines 54 and 55 of page 2)”In reference [5, 6], Irie et al. introduced…” should be revised as “Irie et al. [5,6] introduced…”. Line 215 of page 6 also has the same problem.

7.(Line 143 of page 3)”…16bit data…” should be revised as “”…16-bit data…”.

8.(In page 4)The abbreviations for the x and y labels are not correct in Fig. 1(b).

9.(Line 171 of page 4) The two terms f(x,y) and fs(x,y) have been defined in line 158.

10.(Page 6)There is an uncertain symbol between B and e^T. Line 278 of page 9 also has the same problem.

11.(Page 7) The left brace in eq.(13) should be removed.

12.(Line 287 of page 9) The symbol “η” should be replaced by “n” for eq.(20).

13.(Line 288 of page 9)” …the computed noise total variance of the i-th block.” should be revised as “the estimated noise variance of the i-th block.”.  

14.(Line 302 of page 10) The abbreviation HLMAD is not defined.

15.(Pages 10 and 11) In Algorithms 1 and 2, the line numbers should be revised as step numbers.   

16.(Line 315 of page 10)”…the local mean-total variance pair,” should be revised as “…the local mean-variance pair,”.

Reviewer 3 Report

Dear Editors, Authors,

The presented paper shows interesting method of selection of homogenous blocks for estimation of noise characteristics of CCD/CMOS cameras. The authors show an extensive background and the experiment is prepared the right way. I am not satisfied with the descriptions of the method itself. Even if I can figure out the authors’ approach, it is presented in a confusing way with formulas probably too complicated. The second thing is that the last experiment is in my opinion not necessary. This is enogh to show that the method estimates the gain and readout noise more precise that other methods do. There is no need to show the results of filtration using this method, as it is quite obvious that more accurate estimates will lead to better denoising performance. Cutting this part will shorten the paper, so that it will not be so long. Here goes my next point, that the paper is too long and I wish author consider reduction to make it more condensed. Finally, the English is very poor – it is necessary to ask native to correct the paper. It is unacceptable to publish the paper where the English is at such low level. Please also see the points below:

1.      Is He-Ne laser stabilized well enough to state that the scene remains stable and all you observe is only due to the noise in image sensor ? Please estimate this level of fluctuation and compare with e.g. readout noise.

2.      Figure 1B – please clarify how the estimation of readout noise and gain of camera was performed. It was not given clearly enough in the text. You even didn’t noticed that you estimate gain and read-out noise. Based on the gain, you can estimate the shot noise.

3.      In formula 5, alpha is exactly the gain of camera – please do not explain it as some coefficient – it is a real-world parameter of a sensor.

4.      In formula 5, f_s (with over-line), is not a local mean intensity – it is the expected intensity of a given pixel !

5.      In figure 2 please add 3 examples with only noise to show that the parameter gives smaller values than for textures

6.      Part 3.1.2 is the most important and its clarity should be improved much – in current form it is unacceptable and confusing. Please improve and simplify the formulas here.

Author Response

Round  2

Reviewer 2 Report

The quality of the revised paper has been much improved by the authors. I have no further comments and think this paper can be accepted for publication.

Reviewer 3 Report

Dear Authors, Editors,

In current form the paper is acceptable. Thank you for improvements.